# Influence of perch provision during rearing on activity and musculoskeletal health of pullets

**Mallory G. Anderson**[1], **Alexa M. Johnson**[1], **Cerano Harrison**[1,2], **Jeryl Jones**[1,2], **Ahmed Ali**[1,3] *

**1** Department of Animal and Veterinary Sciences, Clemson University, Clemson, SC, United States of America, **2** South Carolina Translational Research Improving Musculoskeletal Health Center, Clemson, SC, United States of America, **3** Animal Behavior and Management, Veterinary Medicine, Cairo University, Cairo, Egypt

* ali9@clemson.edu

**Data Availability Statement:** https://osf.io/qgm8x/?view_only=18c754a558294c91b92c40e4d84d9a10 Ali, A. (2024, April 23). Influence of perch provision

## Abstract

Prior research suggests exercise during pullet rearing can mitigate lay-phase bone fractures by strengthening muscles, enhancing balance, and increasing bone mass. This study aimed to confirm that Hy-Line brown pullets with multi-tier perches show increased activity and improved musculoskeletal health. Pullets (n = 810) were randomly allocated to housing systems, either with multi-tier perches (P; n = 15 pens) or without (NP; n = 15 pens), spanning from 0–17 weeks of age. At 5, 11, and 17 weeks, individual birds were meticulously monitored for activity using accelerometers over three consecutive days (n = 90 randomly selected birds/week). At 11 and 17 weeks, 60 birds underwent euthanasia and computed tomography (CT) scans to ascertain tibiotarsal bone mineral density and cross-sectional area measurements. Post-CT scanning, birds were dissected for muscle size, tibiotarsal breaking strength, and tibiotarsal ash percentage measurements. Additionally, serum concentrations of bone-specific alkaline phosphatase and procollagen type 1 N-terminal propeptide were assessed as markers of bone formation (n = 90 birds/week). Pullet group P exhibited heightened vertical activity (P<0.05), with no discernible differences in overall activity (P>0.05) during weeks 5, 11, and 17 compared to group NP. Tibiotarsal bones of P pullets demonstrated superior total and cortical bone mineral density at week 11, alongside increased cortical bone cross-sectional areas and heightened total and cortical bone mineral densities at week 17 (P<0.05) compared to NP pullets. At week 11, P pullets displayed larger leg muscles, including triceps, pectoralis major and minor, and leg muscles at week 17 (P<0.05) compared to NP pullets. Notably, at both weeks, P pullets' tibiae exhibited greater breaking strengths, higher ash percentages, and elevated concentrations of bone-specific alkaline phosphatase and procollagen type 1 N-terminal propeptide compared to NP pullets (P<0.05). The study findings underscore the benefits of providing multi-tier perches for pullets, serving as a valuable tool for enhancing bird activity and musculoskeletal health preceding the lay phase.

during rearing on activity and musculoskeletal health of pullets. Retrieved from osf.io/qgm8x.

**Funding:** The author(s) received no specific funding for this work.

**Competing interests:** The authors have declared that no competing interests exist.

## Introduction

Laying hens experience pronounced biological stress to meet the calcium requirements for eggshell formation. To prepare for this impending calcium demand, the function of osteoblasts changes from forming cortical (structural) bone to depositing medullary bone within the cortical bone (mainly in the long bones, such as the femur and tibia) in pullets nearing sexual maturity, due to a surge in estrogen hormone [1, 2]. Medullary bone provides little structural support and is intended as a reliable source of calcium for eggshell formation, and the amount of medullary bone builds up rapidly during the early stages of lay [1, 3–5]. The supply of medullary bone can be replenished by dietary calcium whereas cortical bone cannot, except if the amount of estrogen decreases and egg production ceases [2, 6, 7]. Osteoclasts mobilize calcium for eggshell formation mainly from medullary bone, but will take calcium from cortical bone where medullary bone is thin [8, 9]. Over time, mobilization of the medullary and unreplenishable cortical bone can cause osteoporosis especially during the late laying period, which raises a major animal welfare problem in the laying hen industry [10–14].

Housing systems with perches and greater freedom of movement may offer a potential solution to the negative effects of osteoporosis by providing opportunities for exercise during rearing. Pullets that frequently perform exercise-related activities may be better prepared for the strenuous requirements of the lay phase through improved musculoskeletal health at an earlier age. Bone is typically strengthened when weight-bearing load is applied or when the bone is strained by muscle contractions to induce bone remodeling [15, 16]. Perches can increase activity and load-bearing exercise, as pullets are highly motivated to perch on elevated surfaces [17, 18]. A few previous studies document the beneficial effect of perches on pullet musculoskeletal health. For example, bone mineral content and leg muscle weights were greater in 12-week-old White Leghorn pullets reared with perches compared to those without, indicating exercise via perch use had a beneficial impact on bone mineralization and muscle deposition [19]. Aviary-reared LSL-Lite pullets with access to perches had improved muscle weights and better bone quality than conventional cage-reared pullets at 16 weeks of age [20]. Furthermore, structural bone density of the humeri and tibiae of aviary-reared pullets were greater than for conventional cage-reared White Leghorn pullets, with the former having stronger humeri [21]. Conversely, inactivity has shown to increase the incidence of osteoporosis [10]. However, the literature lacks information on how multi-tier perches may affect the musculoskeletal health of brown-feathered pullet strains. By providing a structure for perching that contained three levels, birds would be able to perform behaviors such as wing-flapping, walking, running, and jumping which would strengthen their musculoskeletal system. The incorporation of a multi-tier perch particularly in earlier age may encourage birds to perch, practice their balance, and jump up from one rung to reach the next, undergoing more strenuous exercise compared to a single-level perch. By increasing vertical activity from loading and unloading exercises associated with perching, we hypothesized that pullets with access to multi-tier perches would experience improved bone density, mineralization, strength, and muscle deposition at 11 and 17 weeks of age, possibly reducing the incidence of osteoporosis or bone fracture later in life. The primary objectives encompassed the comparative analysis of musculoskeletal health metrics (i.e., tibial bone mineral density and cross-sectional area, muscle deposition, tibial breaking strength, and tibial ash percentage) in brown-feathered pullets accommodated with or without multi-tier perches. Additionally, we endeavored to assess pullet activity levels utilizing a body-worn accelerometer and explore markers of bone formation (i.e., bone specific alkaline phosphatase and procollagen type 1 N-terminal propeptide). These biomarkers, not previously examined, afford a distinctive perspective, contributing to a comprehensive understanding of the musculoskeletal health and activity profile of Hy-Line brown pullets.

## Materials and methods

### Ethics

This experiment was approved by and conducted in accordance with requirements of the Clemson University Institutional Animal Care and Use Committee (protocol #: AUP2021-0068).

### Animal and housing

This experiment was conducted in a ventilation- and temperature-controlled poultry house at the Morgan Poultry Center, Clemson, South Carolina, USA, from December 2021 to March 2022. Day-old Hy-Line brown chicks (n = 810) were randomly allocated across 30 pens (27 birds/pen) until 17 weeks of age. Pens (5.2m$^2$) contained 7.6cm of clean pine wood shavings as bedding. From 0 to 3 weeks of age, feed was provided in tube feeders and water in gallon drinkers, and for the first week of life, supplementary feed trays were provided. After 3 weeks, feed was provided in circular adjustable hanging feeders, and water was available in automatic cup drinkers. Feed and water were provided ad libitum. For the first 3 weeks of age, heat was provided by one focal electric brooder per pen and a gas-fired brooder for the entire house. The temperature was initially set at 35-36°C at day 0, then reduced by 2-3°C every week until 3 weeks of age when brooders were removed. Temperature was reduced weekly until 6 weeks of age to 21°C, then maintained until the end of the study, following the standard breed guidelines [22]. The light was provided by one 60-watt incandescent overhead lightbulb per pen and each pen was kept on a decreasing light schedule starting at 20L:4D during the first week and was decreased by increments of either 1.5 or 3 hours until 10L:14D from 7 weeks of age until the end of the study when birds were 17 weeks old [22].

### Treatments

From 0 to 17 weeks of age, 15 pens were provided with perches while the remaining 15 pens were without perches. This resulted in two treatment groups: perch (P) and no perch (NP). The perch structure was constructed to be adjustable with perch rungs made of 5×5cm pressure-treated wooden lumber. Each perch structure contained 3 rungs of varying height, each 165.1cm in length, resulting in 495.3cm of total perch space and approximately 19cm of perch space per bird. In the P group, rung heights and distance between rungs were gradually increased concurrently with the growth of the birds to ensure they were easily accessible. For the first 11 days of age, the 3 rungs were 15.2cm, 22.8cm, and 30.4cm high off the ground. For the next 8 days, the 3 rungs were 22.8cm, 38.1cm, and 54.6cm high off the ground. The perch rungs were altered once more on day 19 of age to 38.1cm, 62.2cm, and 88.4cm high, with a 12.7cm distance between each perch rung.

### Activity

Bird activity was observed continuously over a span of three consecutive days during weeks 5, 11, and 17 of age, with a total of 90 birds per week. At each designated time point, three birds per pen were captured following the cessation of lighting. Birds were selected from various resources and perch heights to ensure a representative sampling of hens from each pen. These birds were equipped with harnesses to which accelerometers were attached. The accelerometers used in this study were 58 × 33 × 23mm in dimensions and weighed 16 grams, with a measuring range of ±3g; 29.4m/s$^2$ and an accuracy level of ±0.105g; 1.03m/s$^2$ when operating within a temperature range of -20°C to 70°C. The orientation of the loggers on the hens allowed for the capture of craniocaudal movement on the X-axis, mediolateral movement on

the Y-axis, and dorsoventral movement on the Z-axis. The loggers were securely fastened within the harnesses to minimize data noise resulting from logger movement and to maintain consistent logger orientation. Following the attachment of harnesses and accelerometers to focal birds, a one-day period was provided for habituation. During this period, close monitoring ensured that the vests did not impact the behavior or locomotion abilities of the hens. After acclimation, the loggers recorded the hens' movements over three consecutive days (72 hours) at each time point, with a scanning frequency of 20 Hz (-3g to +3g) across three axes.

## Computed tomography (CT) image acquisition

At 11 and 17 weeks of age, two birds per pen per week (totaling n = 60) were euthanized on the farm via $CO_2$ inhalation. Following euthanasia, the birds were placed in an ice-cooled cooler and promptly transported to the Godley-Snell Research Center situated on Clemson University's campus. Upon arrival, each bird was individually positioned within a V-shaped foam cradle in a dorsal recumbent posture on a hydroxyapatite calibration phantom (QRM Quality Assurance in Radiology and Medicine, Möhrendorf Germany). The bird's head and legs were extended in opposite directions and secured with tape to maintain this alignment during image acquisition. Computed tomography (CT) images were obtained using a helical mode and a head 0-10kg protocol, with a slice thickness of 0.5mm and utilizing bone and soft tissue reconstruction algorithms. The CT imaging was performed using a Toshiba Aquilion TSX-101A, 16-slice scanner (GE Healthcare, Chicago IL, USA), as described in [23]. Following CT scanning, the birds were promptly dissected and frozen at -29˚C for subsequent analysis.

## Tibiotarsal CT image analysis

For each CT study, measurements of the right tibiotarsal bone and muscle were made using a standardized CT image analysis protocol previously published by [23]. The density (measured in Hounsfield Units, HU) and area (in millimeters) of both the total and medullary components of the tibiotarsal bone were determined at predetermined proximal, middle, and distal transverse slice positions through hand-traced regions of interest. Additionally, the cross-sectional area (CSA) of the muscle group surrounding the tibiotarsus was measured at each of these predefined locations. To calibrate the bone density measurements, the CT densities of each rod in the bone calibration phantom were recorded using the oval ROI tool. Subsequently, the recorded CT densities in HU were converted into hydroxyapatite values using graphical analysis techniques described in [23].

## Muscle deposition

Birds were removed from a –29˚C freezer and allowed to thaw at refrigerated temperature for approximately 48 hours before dissection. The separation of muscles followed procedures described by [20] and with the assistance of a veterinarian (A.A.) to ensure consistent muscle specimen collection. The birds were dissected by making an incision in the skin below the keel bone and carefully peeling it back to expose the bird's interior. The right biceps and triceps brachii were then removed by peeling back the skin of the wing and conducting a blunt dissection along the boundary between the biceps and triceps. Subsequently, the biceps and triceps were delicately separated from the bone, and their proximal and distal tendons were cut at the level of the bone. The pectoralis muscles were separated by cutting the fascia along the delineation line, isolating the fats from the pectoralis muscles, and severing all attachments at the origin (Crania sternum, furcula, and sternal ribs) and at the insertion points of the major (proximal ventral surface of the humerus) and minor (proximal dorsal surface of the humerus). The muscles, tendons, and ligaments of the left leg were detached from the bone,

the Achilles tendon was severed, and the fascia along the synsacrum was removed. Immediately upon removal, all muscles were weighed. The left tibiae were frozen at -29˚C for ash percentage measurements, and the right legs were frozen at -20˚C for later dissection of the tibiae for breaking strength measures.

## Breaking strength

The mechanical characteristics of the right tibiotarsi were evaluated through a three-point bending test, adhering to the guidelines outlined by the American National Standards Institute (ANSI) for the application of three-point bending on animal bones [24]. The tetsing was conducted using an Instron Dynamic and Static Material Test system (Model 5944, Instron Corp., Canton, MA, USA) fitted with a 500N load cell and Automated Material Test System software. Before the commencement of testing, the legs, previously frozen, were allowed to thaw at refrigerator temperature. The surrounding muscles of the tibiotarsus were meticulously dissected, and the length and diameter of the tibiotarsus at the midpoint were documented. Subsequently, the bones were enveloped in saline-soaked paper towels until the commencement of testing. Rounded support pins and breaking blades were manufactured based on ANSI/ASAE S459 MAR1992 (R2017) standards for the application of 3-point bending on animal bones [24]. A furculum width of 4cm was used (Fig 1). The selected width did not strictly adhere to the ANSI standards but was chosen based on a collaborative decision among the co-authors. Given the unique anatomy of the laying hen tibiotarsus, a width of 4cm was deemed appropriate to ensure that the tibiotarsus could rest evenly on the furculum, thus distributing

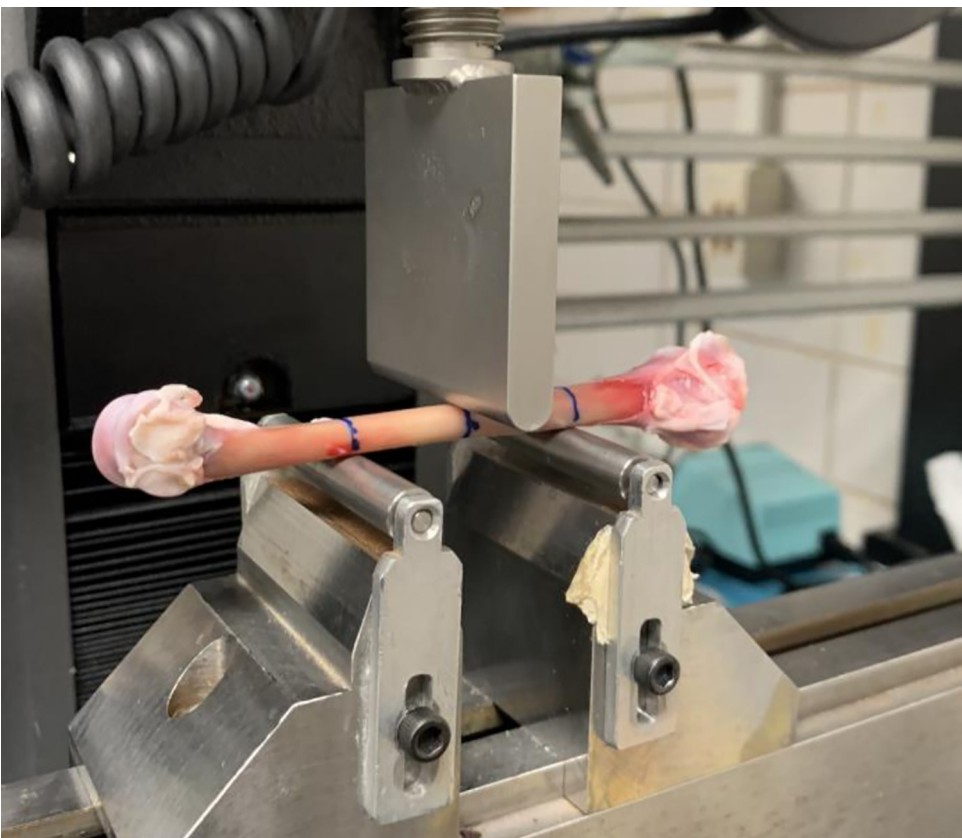

**Fig 1. Instron configuration with rounded supports and breaking blade machined according to ANSI standards.**

the load evenly across the midpoint of the bone in the craniocaudal plane. The testing was conducted at a crosshead speed of 3mm/min, and the test continued until failure occurred. Load and displacement data were gathered and utilized to determine both the breaking strength (N) and stiffness (N/mm).

## Tibia ash percentage

The left tibiae of euthanized birds were thawed for approximately 24 hours prior to the start of data collection. Following thawing, the bones were cleaned of any surrounding muscles and soft tissues, and then separated from the fibula. Subsequently, the tibiae were sectioned into three pieces to ensure they fit properly into a Soxhlet chamber for ether extraction. Ceramic crucibles were air-dried for one hour and then placed in a desiccator for an additional hour, with the weight of the dried crucibles being recorded. The left tibiae were dried at 100˚C for one hour, followed by placement in a desiccator for another hour, and their weight was recorded thereafter. The tibiae were then transferred into the dried ceramic crucibles and subjected to ashing (using an ashing oven: Thermolyne 30400, Barnstead International, Dubuque, IA, USA) for 6 hours at 600˚C. Following ashing, the resultant ash was left in a desiccator for one hour, and its weight was subsequently recorded. The percentage of tibia ash was determined by dividing the weight of the tibia ash by the weight of the dried tibia and multiplying by 100.

## Bone formation markers

During weeks 11 and 17 of age, blood samples were collected from the brachial wing vein of 3 birds per pen per week (n = 90). Whole blood samples were transferred to 1.5mL Eppendorf tubes, and serum was separated at 6000 rpm for 10 minutes at 4˚C. Serum samples were analyzed for levels of bone-specific alkaline phosphatase (BALP) and procollagen type 1 N-terminal propeptide (P1NP) using commercial ELISA kits Nanjing Jiancheng Institute of Bioengineering (Nanjing, China) and MyBioSource (San Diego, CA, USA), respectively.

## Data processing and statistical analysis

The raw accelerometer data, containing information such as date, time, and corresponding impulses in the X, Y, and Z dimensions, were retrieved from the devices (using HOBOware Graphing & Analysis Software, Bourne, MA, USA) at the conclusion of each three-day observation period. Data pertaining to the hens' vertical movement ($a_z$: dorsoventral movement across vertical levels), horizontal movement ($a_x$: craniocaudal movement within the same vertical level), and lateral movement ($a_y$: mediolateral movement within the same vertical level) during daylight hours were directly obtained from the loggers. The hens' triaxial movement ($A_s$) was computed by summing and averaging the raw movement data as follows:

$$A_s = \sqrt{a_x^2 + a_y^2 + a_z^2}$$

Acceleration data were post-processed using MATLAB (MATLAB and Statistics Toolbox Release 2012, The MathWorks, Inc., Natick, MA, USA). To accurately calculate the incidence of massive acceleration shifts on the vertical (z) axis that represents perching, data were smoothed from noisy components by removing all minor acceleration fluctuations using a loop function.

$$A_i = \frac{1}{3} \sum_{j=i-1}^{i+1} A_j \qquad\qquad A_i' = \begin{cases} \mu, if |A_i - \mu| < t \\ \mu, if |A_i - \mu| \geq t' \end{cases}$$

Data smoothing included the passing of the raw acceleration values ($Aj$) through an asymmetrical 3-point-moving average low-pass filter ($I$ = the middle point in the 3-point-moving average low-pass filter) and through a step function to define thresholds used to remove minor fluctuations ($t$ = threshold values of minor fluctuations, i.e., between 0.001 and 0.043g). After processing data, perching events were recognized by detecting massive shifts in acceleration in the z-axis of activity. To precisely detect acceleration shifts due to perching and define thresholds for minor fluctuations in the z-axis, timestamped videos of birds while perching were obtained and compared with the corresponding activity data. Using the approach enabled us to locate shifts in z-axis acceleration mainly caused by perching and define the threshold cutoff points to remove minor fluctuation.

The data were analyzed using R software (version 3.3.1) along with the "stats" package (R Core Team, 2013). Generalized linear mixed-effects models (GLMMs) were employed to examine the main effects of treatment (P and NP) and bird age (activity at weeks 5, 11, and 17; bone formation markers; tibiotarsal BMD and CSA at weeks 11 and 17) on each variable. These models were constructed using the "lme4" package [25]. In each GLMM, the interaction term between main effects was also evaluated as a fixed effect, while bird ID, pen, and day for activity were considered as random effects. The "Quasibinomial" family was chosen for proportion data (ash %), and the "Poisson" family was used for other data types. Post hoc comparisons were conducted using Tukey's HSD multiple comparison procedure with the "multcomp" package [26]. For proportion data (ash %), the "DHARMa" package was utilized to assess residual distribution and GLMM assumptions, while the Shapiro–Wilk test was applied for normality analysis of the model residuals (i.e., activity (g), breaking strength (N), stiffness (N/mm)). Statistical significance was established at $p < 0.05$. Descriptive statistics were computed using the "psych" package, and data were presented as mean ± standard error of the mean (SEM).

## Results

### Activity

At weeks 5, 11, and 17 of age, pullets housed with perches showed increased vertical activity and average daily vertical displacement per bird compared to pullets housed without perches (Table 1). Furthermore, pullets housed with perches had decreased horizontal activity compared to pullets housed without perches at weeks 5, 11, and 17 of age (Table 1). Overall activity levels did not differ between treatment groups at any week of age ($p > 0.05$; Table 1).

### Musculoskeletal health

Muscle deposition. At week 11 of age, pullets housed with perches had greater leg muscle group relative weights compared to pullets without perches ($p = 0.041$; Fig 2). There were no differences between treatments for biceps brachii, triceps brachii, pectoralis major, or pectoralis minor weights at week 11 of age ($p > 0.05$).

At week 17 of age, pullets housed in P pens had greater relative weights of triceps brachii ($p = 0.041$), pectoralis major ($p = 0.032$), pectoralis minor ($p = 0.039$), and leg muscle group ($p = 0.021$) compared to pullets from NP pens (Fig 3). There were no differences between treatments for bicep brachii weights at week 17 of age ($p > 0.05$).

Bone mineral density (BMD) and cross-sectional area (CSA). At week 11 of age, pullets housed with perches had greater cortical CSA at the proximal section of the tibia and greater total and cortical BMD at all regions, with a tendency for a larger cortical BMD at the proximal section compared to pullets housed without perches (Table 2). At week 17 of age, pullets housed with perches had greater cortical CSA, and total and cortical BMD at all sections of the tibia compared to pullets housed without perches (Table 2).

**Table 1. Activity of pullets housed with perches (P) or no perches (NP) for 3 consecutive days at weeks 5, 11, and 17 of age (n = 90 birds/week; g = gravitational force; n = number of daily displacements).**

| Parameter | | Overall Activity (g) | Vertical Activity (g) | Horizontal Activity (g) | Daily Vertical Displacement (n) |
|---|---|---|---|---|---|
| Week/Treatment | | | | | |
| Week 5 | Perch (P) | 1.72±0.36 | 0.73±0.15* | 0.99±0.12 | 26.88±6.85* |
| | No Perch (NP) | 1.65±0.33 | 0.15±0.11 | 1.50±0.22* | 5.36±1.66 |
| | P-value | 0.423 | 0.024 | 0.032 | 0.001 |
| Week 11 | Perch (P) | 1.43±0.32 | 0.61±0.16* | 0.82±0.18 | 15.96±5.85* |
| | No Perch (NP) | 1.38±0.41 | 0.16±0.12 | 1.22±0.32* | 3.22±1.96 |
| | P-value | 0.355 | 0.019 | 0.035 | 0.001 |
| Week 17 | Perch (P) | 1.36±0.35 | 0.52±0.13* | 0.84±0.11 | 13.85±4.52* |
| | No Perch (NP) | 1.29±0.38 | 0.11±0.09 | 1.18±0.19* | 2.52±1.01 |
| | P-value | 0.256 | 0.011 | 0.021 | 0.001 |
| P-value | Week | 0.219 | 0.153 | 0.185 | 0.287 |
| | Treatment | 0.426 | 0.003 | 0.001 | 0.002 |
| | Week × Treatment | 0.328 | 0.001 | 0.001 | 0.001 |

*Means within the same column (parameter), week of age, and across rows (treatments) indicate statistically significant differences (P < 0.05).

Breaking strength. At week 11 of age, pullets housed with perches had greater breaking strength compared to pullets housed without perches (Table 3). At week 17 of age, pullets housed with perches had greater breaking strength and stiffness compared to pullets housed without perches (Table 3). There were no differences between stiffness at week 11 of age (p > 0.05; Table 3).

Tibia ash percentage. At week 11 of age, pullets housed with perches had higher ash percent compared to pullets without perches (Table 4). Similarly, at week 17 of age, pullets housed with perches had higher ash percent compared to pullets without perches (Table 4).

Bone formation markers. During week 11, birds housed in P pens had higher levels of BALP (p = 0.032) and P1NP (p = 0.026) compared to birds housed in NP pens (Figs 4 and 5). Similarly, during week 17, birds housed in P pens had higher levels of BALP (p = 0.011) and

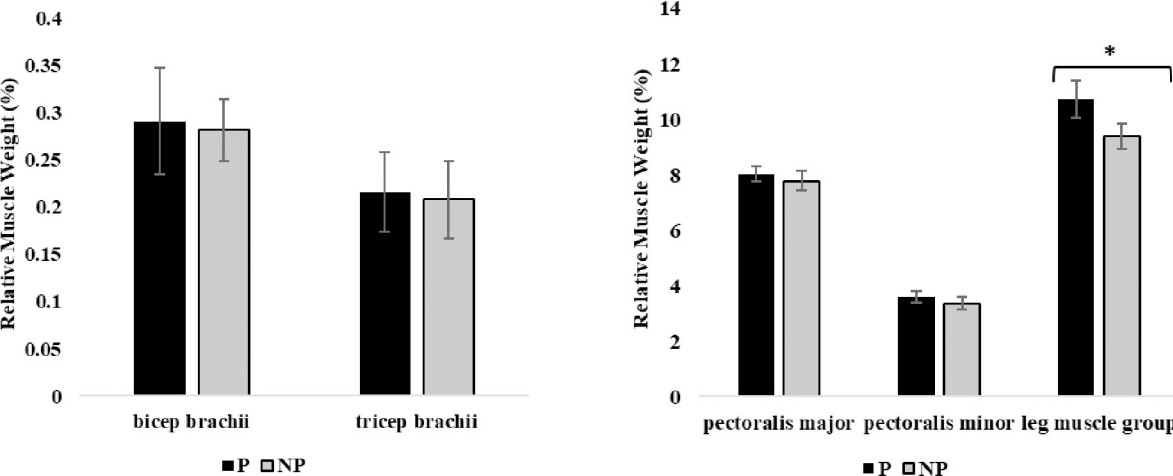

**Fig 2. Relative muscle weight (% of total body weight) of biceps brachii, triceps brachii, pectoralis major, pectoralis minor, and leg muscle group of 11-week-old pullets (n = 60 birds) housed with perches (P) or no perches (NP).** Results are presented as mean relative weight (%) ± SEM. *Across bars indicates significant statistical differences at p < 0.05.

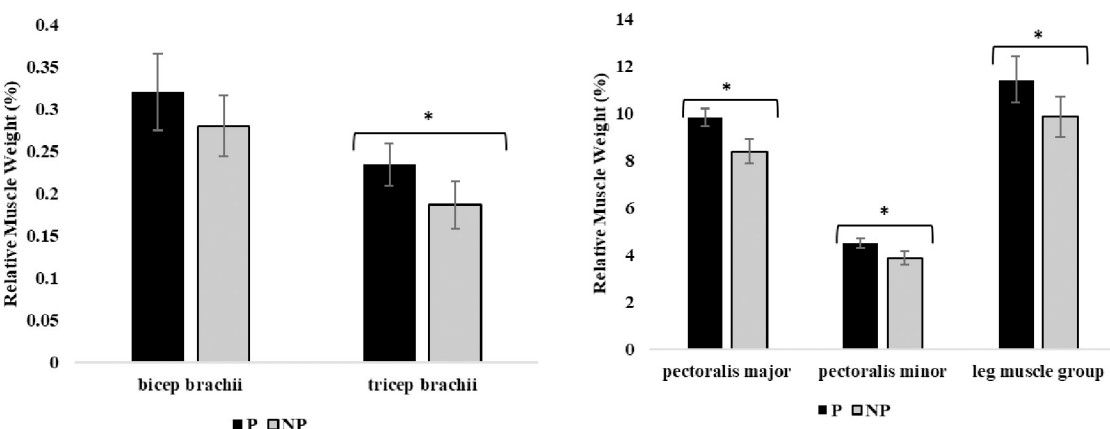

**Fig 3. Relative muscle weight (% of total body weight) of bicep brachii, triceps brachii, pectoralis major, pectoralis minor, and leg muscle group of 17-week-old pullets (n = 60 birds) housed with perches (P) or no perches (NP).** Results are presented as mean relative weight (%) ± SEM. *Across bars indicates significant statistical differences at $p < 0.05$.

**Table 2. Bone mineral density (BMD; mg/cm$^3$) and bone cross-sectional area (CSA; mm$^2$) ± SEM for the total and cortical regions of the right tibiotarsus of pullets housed with perches (P) or no perches (NP) at 11 (n = 60 birds) and 17 (n = 60 birds) weeks of age.**

| Parameter | | Bone Cross-Sectional Area (mm²) | | | | | |
|---|---|---|---|---|---|---|---|
| Week/Treatment | | Total | | | Cortical | | |
| | | Proximal | Middle | Distal | Proximal | Middle | Distal |
| Week 11 | Perch (P) | 59.69±0.96 | 45.86±1.03 | 47.85±1.15 | 40.30±1.03 | 29.46±1.15 | 30.76±1.16 |
| | No Perch (NP) | 58.95±1.02 | 45.36±1.12 | 47.03±1.16 | 37.96±1.55 | 28.19±1.12 | 29.06±1.03 |
| | P-value | 0.351 | 0.152 | 0.216 | 0.043 | 0.253 | 0.152 |
| Week 17 | Perch (P) | 62.69±1.06 | 47.99±0.79 | 49.58±0.88 | 44.64±1.58 | 35.98±1.23 | 36.07±1.03 |
| | No Perch (NP) | 61.52±0.75 | 46.13±0.88 | 48.63±1.01 | 40.99±1.16 | 33.69±1.03 | 34.26±1.81 |
| | P-value | 0.152 | 0.143 | 0.215 | 0.036 | 0.041 | 0.039 |
| P-value | Week | 0.096 | 0.263 | 0.199 | 0.423 | 0.039 | 0.096 |
| | Treatment | 0.185 | 0.258 | 0.452 | 0.023 | 0.046 | 0.036 |
| | Week×Treatment | 0.253 | 0.326 | 0.235 | 0.044 | 0.044 | 0.043 |
| Parameter | | Bone Mineral Density (mg/cm³) | | | | | |
| Week/Treatment | | Total | | | Cortical | | |
| | | Proximal | Middle | Distal | Proximal | Middle | Distal |
| Week 11 | Perch (P) | 340.84±19.17 | 420.63 ±27.76 | 304.92 ±23.10 | 372.10±20.93 | 449.70 ±29.68 | 352.85±26.73 |
| | No Perch (NP) | 276.25±19.89 | 321.82 ±24.86 | 243.13 ±20.97 | 324.73±15.83 | 383.31 ±21.92 | 308.65±20.26 |
| | P-value | 0.002 | 0.012 | 0.031 | 0.093 | 0.039 | 0.044 |
| Week 17 | Perch (P) | 599.68±40.34 | 859.55 ±67.00 | 938.19 ±71.47 | 1196.28±134.58 | 2055.04 ±271.27 | 1651.45±250.19 |
| | No Perch (NP) | 537.88±21.07 | 761.35 ±20.04 | 707.66 ±24.17 | 953.69±107.29 | 1324.22 ±174.80 | 1103.99±167.25 |
| | P-value | 0.011 | 0.012 | 0.021 | 0.013 | 0.011 | 0.011 |
| P-value | Week | 0.013 | 0.014 | 0.036 | 0.026 | 0.023 | 0.034 |
| | Treatment | 0.011 | 0.021 | 0.021 | 0.018 | 0.012 | 0.021 |
| | Week×Treatment | 0.021 | 0.011 | 0.011 | 0.016 | 0.011 | 0.013 |

**Table 3. Breaking strength (N) and stiffness (N/mm) of pullets housed with perches (P) or no perches (NP) at weeks 11 (n = 60 birds) and 17 (n = 60 birds) of age.**

| Parameter | | Breaking strength (N) | Stiffness (N/mm) |
|---|---|---|---|
| **Week/Treatment** | | | |
| Week 11 | Perch (P) | 171.71±9.89 | 184.98±8.67 |
| | No Perch (NP) | 153.05±3.78 | 178.16±13.26 |
| | P-value | 0.021 | 0.103 |
| Week 17 | Perch (P) | 276.63±10.31 | 289.96±12.45 |
| | No Perch (NP) | 220.02±8.14 | 240.59±18.95 |
| | P-value | 0.031 | 0.029 |
| P-value | Week | 0.523 | 0.031 |
| | Treatment | 0.029 | 0.041 |
| | Week × Treatment | 0.017 | 0.038 |

*Means within the same column (parameter), week of age, and across rows (treatments) indicate statistically significant differences ($P < 0.05$).

P1NP (p = 0.016) than birds housed in NP pens (Figs 4 and 5). There were no differences in treatments between weeks 11 and 17 (p = 0.542; Figs 4 and 5).

## Discussion

The objective of this study was to investigate the effects of access to a multi-tier perch during rearing on Hy-Line brown pullet activity and musculoskeletal health. Our results suggest that access to perches during rearing increases vertical activity levels and improves aspects of musculoskeletal health, which may benefit pullets as they enter the lay phase.

### Activity

The addition of multi-tier perches to a floor pen environment increased the vertical activity of pullets, as well as the average daily vertical displacement per bird at weeks 5, 11, and 17 of age. The increase in vertical activity level stimulated by the addition of perches is in agreement with previous studies, especially considering that pullets are highly motivated to perch on elevated surfaces [19, 27, 28]. The vertical movement of perching behavior is performed by wing-assisted jumping, which is a form of load-bearing exercise that can strengthen the musculoskeletal system [20, 21]. Wing-assisted exercise involved in perching behavior likely impacted wing bone characteristics. One limitation of the current study is that we did not analyze wing

**Table 4. Tibia ash percent (%) of pullets housed with perches (P) or no perches (NP) at weeks 11 (n = 30 birds) and 17 (n = 60 birds) of age.**

| Parameter | | Bone Ash (%) |
|---|---|---|
| **Week/Treatment** | | |
| Week 11 | Perch (P) | 54.36±0.12 |
| | No Perch (NP) | 53.98±0.17 |
| | P-value | 0.039 |
| Week 17 | Perch (P) | 54.96±0.21 |
| | No Perch (NP) | 54.19±0.19 |
| | P-value | 0.023 |
| P-value | Week | 0.089 |
| | Treatment | 0.031 |
| | Week × Treatment | 0.022 |

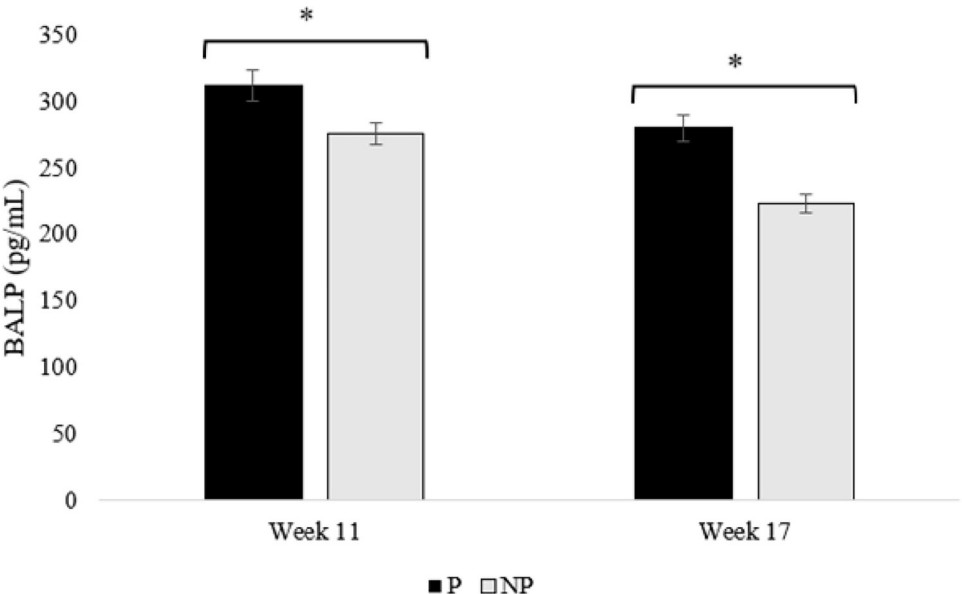

**Fig 4. Concentrations of bone-specific alkaline phosphatase (BALP) for pullets housed in perch (P) and no perch (NP) housing environments during weeks 11 and 17 (n = 90 birds/week).** Results are presented as mean ± SEM. *Across bars indicates significant statistical differences at p < 0.05.

bone strength, such as the humerus, radius, or ulna. Although overall activity levels did not differ between treatment groups, the increase in vertical activity seen in pullets reared with perches suggests they were reaching higher areas of the pen and performing load-bearing exercises more often compared to pullets without perches, which could improve their musculoskeletal health.

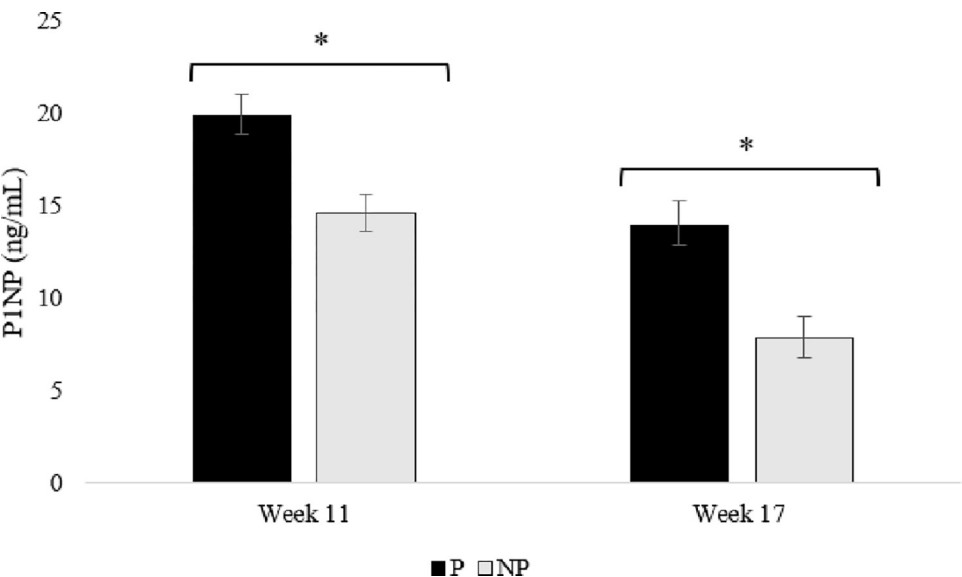

**Fig 5. Concentrations of pro-collagen type 1 n-terminal propeptide (P1NP) for pullets housed in perch (P) and no perch (NP) housing environments during weeks 11 and 17 (n = 90 birds/week).** Results are presented as mean ± SEM. *Across bars indicates significant statistical differences at p < 0.05.

## Muscle deposition

We observed greater leg muscle group relative weights of pullets reared with perches at 11 weeks of age with no differences in the weights of other muscles, suggesting they were engaging the leg muscles more than the breast or wing muscles. In agreement, previous research found that when averaging muscle weights between 3, 6, and 12-week-old pullets, the only observed difference between pullets reared in conventional cages with perches and those without was between thigh, not breast muscle weights [19]. At 17 weeks of age, we observed greater triceps brachii, pectoralis major and minor, and leg muscle group weights in pullets housed with perches than those without. By 17 weeks of age, pullets were likely using their wings to assist in jumping on and off perches, whereas pullets without perches had no such opportunity to engage the wing and breast muscles. Previous research reports that wing, breast, and leg muscle weights differ between aviary-reared and conventional cage-reared pullets at 16 weeks of age [20]. However, one previous study did not find a difference in pectoralis major or minor, bicep, or leg muscle group weights between pullets housed in an open-concept barn with platforms, ramps, and at least six perches and those housed in a single level wire-floor brooding compartment with only two perches at 10 and 16 weeks of age [29]. Interestingly, this previous study found that brown-feathered strains had lower pectoralis major weights, but higher leg muscle group weights compared to white-feathered strains regardless of housing type [29]. The brown-feathered birds used their leg muscles more, resulting in increased leg muscle weights, whereas the white-feathered birds performed more wing-associated behaviors, using their pectoral muscles more than brown-feathered strains, resulting in the increased pectoral weights [29]. Ultimately, muscle development does seem to take time and depend on genetic makeup, where first the leg muscles are mostly engaged theoretically to jump on perches at 11 weeks of age, and second, the wing and breast muscles are also engaged to assist in load bearing exercise associated with jumping on and off the elevated perch rungs.

## Bone mineral density (BMD) and cross-sectional area (CSA)

At week 11 of age, pullets housed with perches had greater cortical CSA at the proximal section of the tibia, and a greater total and cortical BMD compared to pullets housed without perches. Furthermore, at week 17 of age, pullets housed with perches had greater cortical CSA at all regions of the tibia, and greater total and cortical BMD compared to pullets housed without perches. These results indicate that at 11 and 17 weeks of age, pullets reared with perches showed improved bone mass compared to those without. In our study, pullets with access to perches also exhibited more vertical activity and vertical displacement (i.e., jumping) behavior at 5, 11, and 17 weeks of age compared to pullets without access to perches, likely resulting in the beneficial effect observed on BMD and CSA. This is because load bearing exercise associated with perching can positively impact bone development [30, 31]. Our findings in brown-feathered pullets support previous literature, where white-feathered pullets housed in furnished cages with platforms and terraces had higher bone mineral densities than pullets in conventional cages at 4, 12, and 16 weeks of age [32]. Furthermore, pullets provided opportunities for load bearing exercise in the form of wing-assisted jumping and increased vertical activity showed higher bone mineral density compared to pullets without opportunity to perform such exercise [20, 21, 33–35]. Furthermore, 16-week-old pullets reared in an aviary system showed thicker cortices in the tibia and humerus compared to those reared in conventional cages [21]. Our findings indicate a higher amount of structural bone in pullets housed with perches. This greater amount of total and cortical bone density and area resulting from the perch treatment will likely benefit pullets as they reach the lay phase compared to pullets reared without perches. As pullets enter the lay phase, osteoclasts mobilize calcium from

the cortical and medullary bone to be used for eggshell formation. Over time, this prolonged loss of nutrients from the bones results in weakness and susceptibility to fracture. By having a large cortical bone density and area (indicating strong bones) before the start of the lay phase, pullets may be less prone to fracture later on in their adult life. Indeed, it has been indicated that providing perches to pullets can have a long-term beneficial impact on musculoskeletal health of adult laying hens [36–38]

## Breaking strength

Pullets housed with perches showed greater breaking strength at 11 and 17 weeks of age, and greater stiffness at 17 weeks of age compared to pullets housed without perches. The greater breaking strength observed at both testing weeks indicates that pullets reared with perches had stronger bones as early as 11 weeks of age compared to those without perches, which is in line with our activity, muscle deposition, BMD, and CSA findings. Also in agreement, previous studies demonstrate that the force required to fracture the humerus and tibia of aviary-reared pullets is higher than for conventional cage-reared pullets [20, 39]. Additionally, pullets reared in an open-concept barn with platforms, ramps, and at least six perches had stronger tibiae and femurs compared to pullets in in a single level wire-floor brooding compartment with only two perches at 10 and 16 weeks of age [29]. Considering that pullets housed with multi-tier perches were performing more vertical activity and vertical displacement per day, it follows that their tibiae would be stronger than pullets without access to multi-tier perches. In pullets reared with perches, we observed a greater stiffness (rigidity) at 17 weeks of age compared to pullets reared without perches. Previous research discovered similar results, where 16-week-old pullets reared in aviaries had higher stiffness values compared to pullets reared in conventional cages [21]. Bone is a complex material, and its strength and health stems from the delicate balance between rigidity and elasticity [40]. The bone must be stiff (rigid) enough to withstand applied force and allow for load-bearing exercise, but also elastic and flexible enough to absorb energy [40–42]. Therefore, these interplaying variables indicate a tibia that is more resistant to fracture and strong enough to facilitate complex locomotion (i.e., jumping to and from perch rungs of varying heigh and distance). Ultimately, the bone breaking strength and stiffness measures indicate that rearing pullets with perches beneficially alters bone composition so that the overall bone is stronger and the force required to fracture increases.

## Tibia ash percentage

At weeks 11 and 17 of age, pullets housed with perches had a higher ash percent compared to pullets without perches, suggesting improved bone quality which is also in agreement with and reflected by our previous measures of musculoskeletal health. Ash percent is used to measure the amount of minerals in the bone, translating to overall bone health, and has been highly correlated to quantitative computed tomography calculated tibial bone mineral content in laying hens [43]. One previous study found no difference in tibia ash percent between 16-week-old White Leghorn pullets housed in aviary or conventional cage systems, but they did find differences in humerus ash percent, indicating the tibia and humerus respond differently to load-bearing exercise during development [21]. As previously discussed, our brown-feathered pullets reared with perches had heavier leg muscle relative weights in relation to total body weight than those without access to perches, so it does track that the tibia ash percent would be greater in the group of pullets with access to perches than those without due to increased activity levels. Although we did not analyze humerus ash percent, the significant difference in tibia ash percent suggests that providing multi-tier perches during development does improve tibiotarsal bone mineral content through increased vertical activity.

### Bone formation markers

We observed higher levels of BALP and P1NP in birds reared with perches compared to those without perches at weeks 11 and 17 of age. Although a measure of bone mineralization in poultry, higher concentrations of BALP and P1NP can indicate greater rates of bone mineralization, as they are both markers of bone formation [44, 45]. BALP is produced by osteoblasts and is a specific marker of bone formation and osteoblast activity [44, 46, 47]. During the secretion of collagen, which forms the basis of the bone matrix, the N-terminal propeptide (P1NP) is cleaved off and indicates bone formation activity [45, 48, 49]. However, abnormally high levels of BALP and P1NP may suggest underlying problems such as bone disease [50]. But, most other measures of bone health and quality were improved in pullets reared with perches, supporting this was not the case. Based on our review of the literature, this is the first study to evaluate the effect of perches on biomarkers of bone formation in pullets and indicates a positive effect of activity associated with perch use on bone formation markers.

## Conclusion

In the current study, we observed significantly elevated levels of vertical locomotor activity, enhanced muscular tissue deposition, increased bone mineral density, improved bone biomechanical characteristics, elevated tibia ash content, and heightened bone mineralization in Hy-Line brown pullets provided with multi-tier perches compared to those deprived of access to perches. These discernible enhancements in pullets suggest that weight-bearing physical activity resulting from interaction with perches exerts a beneficial influence on the musculoskeletal properties of pullets at both 11 and 17 weeks of age. Providing pullets with multi-tier perches from 0 to 17 weeks of age promotes exercise, improves musculoskeletal health, and stimulates vertical activity, subsequently better preparing them for the lay phase and potentially reducing the risk of bone fractures in the future. These findings are in agreement with previous studies in white-feathered strains [20, 21]. Subsequent studies should aim to enhance our understanding of the long-term impacts of perching interventions on pullet welfare and bone health.

## Acknowledgments

We would like to thank all the staff and student workers at the Morgan Poultry Center for their time and effort on this project.

## Author Contributions

**Conceptualization:** Ahmed Ali.

**Data curation:** Mallory G. Anderson, Cerano Harrison.

**Formal analysis:** Cerano Harrison, Ahmed Ali.

**Funding acquisition:** Ahmed Ali.

**Investigation:** Mallory G. Anderson.

**Methodology:** Mallory G. Anderson, Alexa M. Johnson, Jeryl Jones, Ahmed Ali.

**Project administration:** Mallory G. Anderson, Alexa M. Johnson.

**Resources:** Jeryl Jones.

**Software:** Cerano Harrison, Jeryl Jones.

**Supervision:** Jeryl Jones, Ahmed Ali.

**Validation:** Cerano Harrison.

**Writing – original draft:** Mallory G. Anderson, Alexa M. Johnson.

**Writing – review & editing:** Mallory G. Anderson, Ahmed Ali.

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
