## [Decision Letter · Decision Letter 0]

13 Mar 2024

PONE-D-24-03112Influence of perch provision during rearing on activity and musculoskeletal health of pulletsPLOS ONE

Dear Dr. Ali,

Thank you for submitting your manuscript to PLOS ONE. After careful consideration, we feel that it has merit but does not fully meet PLOS ONE’s publication criteria as it currently stands. Therefore, we invite you to submit a revised version of the manuscript that addresses the points raised during the review process. Please submit your revised manuscript by Apr 27 2024 11:59PM. If you will need more time than this to complete your revisions, please reply to this message or contact the journal office at plosone@plos.org. Please include the following items when submitting your revised manuscript:A rebuttal letter that responds to each point raised by the academic editor and reviewer(s). You should upload this letter as a separate file labeled 'Response to Reviewers'.A marked-up copy of your manuscript that highlights changes made to the original version. You should upload this as a separate file labeled 'Revised Manuscript with Track Changes'.An unmarked version of your revised paper without tracked changes. You should upload this as a separate file labeled 'Manuscript'.

We look forward to receiving your revised manuscript.

Kind regards,

Ewa Tomaszewska, DVM Ph.D

Academic Editor

PLOS ONE

Journal Requirements:

https://www.mdpi.com/2076-2615/14/2/265

https://www.sciencedirect.com/science/article/pii/S0032579119579305?via%3Dihub

https://www.frontiersin.org/articles/10.3389/fanim.2023.1258935/full

In your revision ensure you cite all your sources (including your own works), and quote or rephrase any duplicated text outside the methods section. Further consideration is dependent on these concerns being addressed.

Reviewers' comments:

Reviewer's Responses to Questions

**Comments to the Author**

1. Is the manuscript technically sound, and do the data support the conclusions?

Reviewer #1: Partly

2. Has the statistical analysis been performed appropriately and rigorously? 

Reviewer #1: Yes

3. Have the authors made all data underlying the findings in their manuscript fully available?

Reviewer #1: Yes

4. Is the manuscript presented in an intelligible fashion and written in standard English?

Reviewer #1: Yes

5. Review Comments to the Author

Reviewer #1: Having reviewed the previous work of these authors (10.3390/ani14020265), I find my comments to be similar in some respects. Initially, I must address a concern regarding the originality of some images in this manuscript. Specifically, Figure 1 appears identical to Figure 1 from 10.3390/ani14020265, Figure 2 to Figure 1 in 10.2460/ajvr.23.05.0109, and Figure 3 to Figure 2 in 10.3390/ani14020265. This could potentially raise copyright issues, and I request the editorial board to address this concern.

Furthermore, there's a discrepancy between the funding information within the manuscript, listing three funders, and the Financial Disclosure section stating, “The author(s) received no specific funding for this work.” This also requires clarification.

General comments:

Large sections of the methodology are verbatim copies of previous work (10.3390/ani14020265).

The manuscript lacks a clearly defined aim and research hypothesis. It should include definitions for "musculoskeletal health metrics" and "novel biomarkers of bone mineralization” (after reviewing the manuscript, it's completely unclear to me what these are). These definitions should be incorporated into the introduction.

My previous general comments also apply to this study:

Including bone weight and its relative proportion to total body weight could offer valuable insights into the skeletal effects of perch access. Similarly, reporting muscle weight as relative muscle weight would enhance understanding of muscle development among different groups. These are crucial for determining whether perch inclusion indeed had a quantitative effect on musculoskeletal development. Refer to the discussion at L380 – without correlating muscle mass to total body weight, the statement remains inconclusive.

Given the availability of CT scans, presenting cross-section images of the tibiotarsal bone mid-diaphysis could visually illustrate potential changes in medullary bone formation among the hen groups (see discussion, L425).

The study would benefit from recognizing certain limitations, such as not analyzing the wing bone. Perch activity likely affects the bones of the chest, shoulder girdle, and wings, as indicated by changes in pectoral muscle mass. See discussion sections L371-373 and L374-376 for examples.

Specific comments:

Is there a specific rationale, related to birds' development stages or changes in feeding/housing regimes, for selecting these particular time points (5, 11, and 17 weeks of age)? Please provide clarification.

In the first paragraph, please seek more recent references concerning osteoporosis in laying hens. Citing excellent but over twenty-year-old studies suggests that this problem has been known for a quarter-century, yet, despite hundreds of research papers, it remains unresolved. Changes in housing conditions, alternative animal keeping methods, consumer knowledge, dietary recommendations, and legal guidelines have significantly improved animal welfare. Please find more recent works addressing the frequency of bone fracture cases in laying hens.

L116 omitted reference

L160 As previously, please specify the average duration of a CT scan for a single bird. If 60 birds were scanned with an average scan time of 10 minutes, this implies 10 hours of continuous measurements. Was the CT analysis conducted in a single day?

L200 and L209 Clarification is needed. L200 indicates that only tibiae were frozen, while L209 suggests that whole legs were frozen.

L221 I reiterate my previous comment: In a three-point bending test, the maximal bending moment equals FL/4, where F is the breaking strength and L is the span width. Since the span width was consistent across samples (4 cm), the maximum bending moment is merely the breaking strength F multiplied by 0.01 m. Including bending moment results does not introduce new data. Additionally, the correct unit for bending moment is Nm, not N/m, leading to confusion about what was actually calculated. Therefore, bending moment should be removed from the manuscript.

L236 and L345 Consider using the term "bone formation markers" for clarity

L258 After further investigation, I found another article by one of the authors (https://www.frontiersin.org/articles/10.3389/fanim.2023.1258935/full) where the smoothing equations turned out to be not one, but two. Which is correct? Necessary revisions should be made accordingly. The current manuscript's equation appears incorrect, as it results in μ regardless of whether (Ai-mi) is greater or smaller than t (the same applies to the work published in Animals; despite the authors' assurances, the equation is essentially identical in all cases).

Table 1 – change “f “ (frequency) to “n” (number of daily activities)

L398-400 please remove, as such conclusions cannot be drawn without a direct comparison with white-feathered pullets since only brown-feathered strains of hens were studied.

L425 As no medullary bone measurements were performed, the assumption is speculative.

The references are not formatted according to the journal's requirements.

6. PLOS authors have the option to publish the peer review history of their article (what does this mean?). If published, this will include your full peer review and any attached files.

Reviewer #1: No

---

## [Author Response · Author response to Decision Letter 0]

23 Apr 2024

Thank you to the editorial board and the reviewer for providing thoughtful comments to improve the integrity of this manuscript. We have attached our responses to the reviewer comments below.

General comments:

Large sections of the methodology are verbatim copies of previous work (10.3390/ani14020265).

AU: Thank you so much for your guidance, we have revised the methodology section to ensure that it does not contain verbatim copies of our previous work. We have taken your guidance seriously and have made the necessary changes to rephrase the text while still accurately conveying the methodology employed in our study.

The manuscript lacks a clearly defined aim and research hypothesis. It should include definitions for "musculoskeletal health metrics" and "novel biomarkers of bone mineralization” (after reviewing the manuscript, it's completely unclear to me what these are). These definitions should be incorporated into the introduction.

Thank you for pointing this out. The aim of the study is provided in L26-28, as well as in L99-103. We added specific musculoskeletal health metrics within the objective for clarification. The hypothesis can be found in L97-98.

My previous general comments also apply to this study:

Including bone weight and its relative proportion to total body weight could offer valuable insights into the skeletal effects of perch access. Similarly, reporting muscle weight as relative muscle weight would enhance understanding of muscle development among different groups. These are crucial for determining whether perch inclusion indeed had a quantitative effect on musculoskeletal development. Refer to the discussion at L380 – without correlating muscle mass to total body weight, the statement remains inconclusive.

AU: We greatly appreciate your valuable suggestion, which we have taken into careful consideration. In response, we have revised the muscle deposition section of our study, presenting the data as relative muscle weight in relation to body weight, as per your recommendation. Regarding the bone aspect, our intention was not to report bone weight; rather, we aimed to emphasize bone area and mineral density in our analysis.

Given the availability of CT scans, presenting cross-section images of the tibiotarsal bone mid-diaphysis could visually illustrate potential changes in medullary bone formation among the hen groups (see discussion, L425).

AU: thank you so much for such an excellent idea; we tried to include some images; however, since we use the multiplanar reformatting (MPR) feature of Horos to make these measurements, capturing the images was very challenging, and quality was truly impacted which make it difficult to incorporate in the manuscript and publish them. 

The study would benefit from recognizing certain limitations, such as not analyzing the wing bone. Perch activity likely affects the bones of the chest, shoulder girdle, and wings, as indicated by changes in pectoral muscle mass. See discussion sections L371-373 and L374-376 for examples.

This is a great point. Thank you for recognizing this limitation, which has been included within the relevant discussion areas (L404-406).

Specific comments:

Is there a specific rationale, related to birds' development stages or changes in feeding/housing regimes, for selecting these particular time points (5, 11, and 17 weeks of age)? Please provide clarification.

AU: We endeavored to align our study with the developmental stages corresponding to changes in dietary supplementation. Specifically, we designated week 5 as the conclusion of the starter diets phase, week 11 as the conclusion of the grower diet phase, and week 17 as the conclusion of the pre-lay diet phase, in accordance with the Hy-Line International Standard Management Guidelines. 

In the first paragraph, please seek more recent references concerning osteoporosis in laying hens. Citing excellent but over twenty-year-old studies suggests that this problem has been known for a quarter-century, yet, despite hundreds of research papers, it remains unresolved. Changes in housing conditions, alternative animal keeping methods, consumer knowledge, dietary recommendations, and legal guidelines have significantly improved animal welfare. Please find more recent works addressing the frequency of bone fracture cases in laying hens.

Thank you for pointing this out. We have found and included more recent relevant references within this paragraph.

L116 omitted reference

This reference has been amended.

L160 As previously, please specify the average duration of a CT scan for a single bird. If 60 birds were scanned with an average scan time of 10 minutes, this implies 10 hours of continuous measurements. Was the CT analysis conducted in a single day?

AU: CT scanning was conducted for all the hens on the same day, images were acquired and analysis was conducted afterward. 

L200 and L209 Clarification is needed. L200 indicates that only tibiae were frozen, while L209 suggests that whole legs were frozen.

Thank you for catching this error, clarification has been provided in L225-227.

L221 I reiterate my previous comment: In a three-point bending test, the maximal bending moment equals FL/4, where F is the breaking strength and L is the span width. Since the span width was consistent across samples (4 cm), the maximum bending moment is merely the breaking strength F multiplied by 0.01 m. Including bending moment results does not introduce new data. Additionally, the correct unit for bending moment is Nm, not N/m, leading to confusion about what was actually calculated. Therefore, bending moment should be removed from the manuscript.

Thank you for pointing this out. Maximum bending moment has been removed from the relevant areas within the results and discussion.

L236 and L345 Consider using the term "bone formation markers" for clarity

Thank you for this suggestion, it has been applied within the manuscript.

L258 After further investigation, I found another article by one of the authors (https://www.frontiersin.org/articles/10.3389/fanim.2023.1258935/full) where the smoothing equations turned out to be not one, but two. Which is correct? Necessary revisions should be made accordingly. The current manuscript's equation appears incorrect, as it results in μ regardless of whether (Ai-mi) is greater or smaller than t (the same applies to the work published in Animals; despite the authors' assurances, the equation is essentially identical in all cases).

AU: Thank you sincerely for your valuable comments. You are indeed correct; there are two equations in our submission. Unfortunately, during the formatting process, these equations were inadvertently merged together in the version you received. We have since rectified this issue and separated them accordingly. It's important to note that we are employing the same approach previously developed in our lab to assess activity. Specifically, we are examining significant shifts in vertical activity, indicative of vertical displacement, within this study.

Table 1 – change “f “ (frequency) to “n” (number of daily activities)

This has been edited within Table 1.

L398-400 please remove, as such conclusions cannot be drawn without a direct comparison with white-feathered pullets since only brown-feathered strains of hens were studied.

This sentence has been omitted.

L425 As no medullary bone measurements were performed, the assumption is speculative.

AU: Thank you very much for your valuable insights; we wholeheartedly agree. While the medullary bone measurements were not incorporated, we did observe an increase in the density and thickness of the cortical bone. This observation still underscores and emphasizes the impact of the intervention.

The references are not formatted according to the journal's requirements.

Thank you for catching this mistake, the references have been updated to meet the journal requirements.

---

## [Decision Letter · Decision Letter 1]

20 May 2024

PONE-D-24-03112R1Influence of perch provision during rearing on activity and musculoskeletal health of pulletsPLOS ONE

Dear Dr. Ali,

Thank you for submitting your manuscript to PLOS ONE. After careful consideration, we feel that it has merit but does not fully meet PLOS ONE’s publication criteria as it currently stands. Therefore, we invite you to submit a revised version of the manuscript that addresses the points raised during the review process.

We look forward to receiving your revised manuscript.

Kind regards,

Ewa Tomaszewska, DVM Ph.D

Academic Editor

PLOS ONE

Journal Requirements:

Reviewers' comments:

Reviewer's Responses to Questions

**Comments to the Author**

1. If the authors have adequately addressed your comments raised in a previous round of review and you feel that this manuscript is now acceptable for publication, you may indicate that here to bypass the “Comments to the Author” section, enter your conflict of interest statement in the “Confidential to Editor” section, and submit your "Accept" recommendation.

Reviewer #1: (No Response)

2. Is the manuscript technically sound, and do the data support the conclusions?

Reviewer #1: (No Response)

3. Has the statistical analysis been performed appropriately and rigorously? 

Reviewer #1: Yes

4. Have the authors made all data underlying the findings in their manuscript fully available?

Reviewer #1: Yes

5. Is the manuscript presented in an intelligible fashion and written in standard English?

Reviewer #1: Yes

6. Review Comments to the Author

Reviewer #1: Thank you to the authors for the revisions, which have significantly improved the quality of the manuscript and clarified most of the doubts or comments raised during the review. However, I still have reservations about the claim that Bone Alkaline Phosphatase (BALP/BLP) and Procollagen Type 1 N-Terminal Propeptide (P1NP) are "novel markers of bone formation." These are standard markers of bone turnover (in this case, osteolysis) that have been used for several years in both human and veterinary medicine. Commercial ELISA kits for poultry have been available from many suppliers for years, making it difficult to regard these markers as "novel." Also, in the case of laying hens (at various ages), not just broilers, the use of ELISA tests to measure levels of BALP and P1NP in blood serum as markers of osteolysis has been repeatedly employed, as evidenced by numerous publications in the last 10 years. For example, I used these markers in my own research on laying hens five years ago, and even at that time, I did not consider them "novel markers" but rather as a well-established, reliable analytical method in the study of bone metabolism in laying hens. In the form currently presented by the authors in the revision, I see no reason to call these assays"novel."

7. PLOS authors have the option to publish the peer review history of their article (what does this mean?). If published, this will include your full peer review and any attached files.

Reviewer #1: No

---

## [Author Response · Author response to Decision Letter 1]

13 Jun 2024

Reviewer #1: Thank you to the authors for the revisions, which have significantly improved the quality of the manuscript and clarified most of the doubts or comments raised during the review. However, I still have reservations about the claim that Bone Alkaline Phosphatase (BALP/BLP) and Procollagen Type 1 N-Terminal Propeptide (P1NP) are "novel markers of bone formation." These are standard markers of bone turnover (in this case, osteolysis) that have been used for several years in both human and veterinary medicine. Commercial ELISA kits for poultry have been available from many suppliers for years, making it difficult to regard these markers as "novel." Also, in the case of laying hens (at various ages), not just broilers, the use of ELISA tests to measure levels of BALP and P1NP in blood serum as markers of osteolysis has been repeatedly employed, as evidenced by numerous publications in the last 10 years. For example, I used these markers in my own research on laying hens five years ago, and even at that time, I did not consider them "novel markers" but rather as a well-established, reliable analytical method in the study of bone metabolism in laying hens. In the form currently presented by the authors in the revision, I see no reason to call these assays"novel."

AU: Thank you for this comment and we would agree. We have updated the manuscript throughout to reflect this.

---

## [Decision Letter · Decision Letter 2]

1 Jul 2024

Influence of perch provision during rearing on activity and musculoskeletal health of pullets

PONE-D-24-03112R2

Dear Dr. Ahmed Ali,

We’re pleased to inform you that your manuscript has been judged scientifically suitable for publication and will be formally accepted for publication once it meets all outstanding technical requirements.

Kind regards,

Ewa Tomaszewska, DVM Ph.D

Academic Editor

PLOS ONE

Additional Editor Comments (optional):

Reviewers' comments:

Reviewer's Responses to Questions

**Comments to the Author**

1. If the authors have adequately addressed your comments raised in a previous round of review and you feel that this manuscript is now acceptable for publication, you may indicate that here to bypass the “Comments to the Author” section, enter your conflict of interest statement in the “Confidential to Editor” section, and submit your "Accept" recommendation.

Reviewer #1: (No Response)

2. Is the manuscript technically sound, and do the data support the conclusions?

Reviewer #1: (No Response)

3. Has the statistical analysis been performed appropriately and rigorously? 

Reviewer #1: (No Response)

4. Have the authors made all data underlying the findings in their manuscript fully available?

Reviewer #1: (No Response)

5. Is the manuscript presented in an intelligible fashion and written in standard English?

Reviewer #1: (No Response)

6. Review Comments to the Author

Reviewer #1: (No Response)

7. PLOS authors have the option to publish the peer review history of their article (what does this mean?). If published, this will include your full peer review and any attached files.

Reviewer #1: No

---

## [Editor Report · Acceptance letter]

4 Jul 2024

PONE-D-24-03112R2 

PLOS ONE

Dear Dr. Ali, 

I'm pleased to inform you that your manuscript has been deemed suitable for publication in PLOS ONE. Congratulations! Your manuscript is now being handed over to our production team.

Kind regards, 

on behalf of

Professor Ewa Tomaszewska 

Academic Editor

PLOS ONE